# Pharmacodynamics and Pharmacokinetics of a New Type of Compound Lansoprazole Capsule in Gastric Ulcer Rats and Beagle Dogs: Importance of Adjusting Oxidative Stress and Inflammation

**DOI:** 10.3390/pharmaceutics11020049

**Published:** 2019-01-22

**Authors:** Binbin Wei, Yan Wang, Huizhe Wu, Mingyan Liu, Weifan Yao, Minjie Wei

**Affiliations:** School of Pharmacy, China Medical University, Shenyang 110021, China; bbwei@cmu.edu.cn (B.W.); ywang99@cmu.edu.cn (Y.W.); hzwu@cmu.edu.cn (H.W.); myliu@cmu.edu.cn (M.L.); wfyao@cmu.edu.cn (W.Y.)

**Keywords:** Lansoprazole, rapid absorption, UPLC-MS/MS, pharmacodynamics, pharmacokinetics

## Abstract

The aim of this study was to investigate the pharmacodynamics and pharmacokinetics of a new type of compound lansoprazole capsule in gastric ulcer rats and beagle dogs in order to confirm whether it is more effective in treating gastric ulcers and its rapid absorption. A rat model of gastric ulcers was used to evaluate the anti-ulcerogenic effect of the compound lansoprazole capsule. A fast and sensitive UPLC-MS/MS method was developed to detect lansoprazole in dog plasma. Macroscopic and histological evaluation results revealed that the compound lansoprazole capsule is more effective in treating gastric ulcers as it was able to significantly reduce the gastric ulcer compared to the other groups. Additionally, it was able to enhance the expression of the antioxidant enzyme superoxide dismutase (SOD) and suppress lipid peroxidation as indicated by the reduction of malondialdehyde (MDA) and H^+^-K^+^-ATP activity. Furthermore, this capsule increased the expression of mucosal vascular endothelial growth factor (VEGF) and cyclic oxygenase 2 (COX-2). The established UPLC-MS/MS method was successfully applied to the evaluation of pharmacokinetic parameters of lansoprazole in beagle dogs. The results indicate that the compound lansoprazole capsule had an advantage of rapid absorption. This study demonstrated that the compound lansoprazole capsule has better gastroprotective activity and that this might be related to its positive influence on oxidative stress and inflammation. This new type of compound lansoprazole capsule may be potentially useful in preclinical therapy.

## 1. Introduction

Lansoprazole (LSZ), a benzimidazole derivative, is a proton pump inhibitor (PPI) that effectively inhibits gastric acid secretion by irreversibly binding to the proton pump (H^+^-K^+^-ATPase) in gastric parietal cells [1,2,3]. Due to its great effectiveness, LSZ has been widely used in the treatment of various acid-related disorders, such as gastric and duodenal ulcers, reflux esophagitis and Zollinger–Ellison syndrome [4,5,6,7]. The pathology of gastric ulcers generally involves multiple processes, such as oxidative stress, apoptosis and inflammatory response. The metabolism and pharmacokinetics of LSZ have been investigated in the past [8,9,10,11,12].

Since LSZ is acid-labile, it is usually administered in the form of capsules containing enteric-coated granules to prevent gastric decomposition and to increase bioavailability. However, it may be decreased if administered within 30 min of food intake and the peak concentration was found to occur within 1 ± 2 h after oral administration. Therefore, because of its instability and slow-absorption, new types of LSZ preparations should be developed and validated for preclinical treatment [13,14,15,16].

Based on current research [17,18], in China, we combined with Liaoning Yilingkechuang Biomedicine Company to develop a new type of rapidly absorbed compound LSZ capsule (to which was added a certain proportion of NaHCO_3_ to form astomach-soluble preparation), with the merits of extremely rapid absorption and fewer problems with stability. This study was the first to evaluate its potential anti-gastric-ulcer pharmacodynamic effects through the glacial acetic-acid-induced gastric ulcer rat model and to clarify its probable mechanism of action. Furthermore, we wanted to determine whether this new type of rapidly absorbed compound LSZ capsule was a more effective treatment for gastric ulcers compared to the other LSZ products.

As a new type of compound LSZ capsule, there is a lack of pharmacokinetic research in terms of its absorption, distribution and metabolism in animals. Therefore, in our pharmacokinetic study, the objective was to compare two LSZ products (the reference brand enteric-coated capsules and test compound LSZ capsule) and the pharmacokinetic parameters in beagle dogs after they were given low, middle and high dosages for both preparations. It is important to develop a rapid and sensitive method for the determination of LSZ in plasma. This paper describes an efficient UPLC-MS/MS method for the determination of LSZ (Figure 1) in dog plasma and furthermore applied this to determine the pharmacokinetics of two LSZ products in order to investigate the advantages of the new type of rapidly absorbed compound LSZ capsule. It was expected that the study would be useful for improving preclinical therapeutic efficacy and further pharmacological studies of LSZ. The novelty of this research is that it found that the new type of compound LSZ capsule is a more effective treatment for gastric ulcers with rapid absorption.

## 2. Materials and Methods

### 2.1. Chemicals and Materials

Lansoprazole (98%) and omeprazole (OPZ, 98%, internal standard [I.S.]) were purchased from the National Institute for the Control of Pharmaceutical and Biological Products (Beijing, China). The test products, which were30-mg capsule of compound lansoprazole (batch: 160329-1; Liaoning Yilingkechuang Biomedicine Company, Liaoning, China) and reference products, which were 30-mg enteric-coated capsules of lansoprazole (brand name: Tianjinwutian, batch: 147A; Tianjin, China), were selected for this study. Superoxide dismutase (SOD) and malondialdehyde (MDA) were obtained from the National Institute for the Control of Pharmaceutical and Biological Products (Beijing, China).

Methanol and formic acid of HPLC grade were purchased from Fisher Scientific (Fair Lawn, NJ, USA). Ethyl acetate (HPLC grade) was provided by Shandong Yuwang Industrial Co. Ltd. (Yucheng, China).

### 2.2. Pharmacodynamic Study in Rats

#### 2.2.1. Animals

A total of 42 male SD rats (180–220 g body weight) were provided by the Experimental Animal Center of China Medical University. The experimental protocol was approved by the Animal Ethics Committee of China Medical University (Permit number: SYK2017-0021), and all animal studies were carried out in accordance with the Guidelines for Animal Experimentation.

#### 2.2.2. Glacial Acetic-Acid-Induced Gastric Ulcer Model

The 30-mg capsule of compound lansoprazole (LSZ: NaHCO_3_): (30 mg/1.1 g) was equivalent to a rat dose of (2.7 mg/99 mg). The rats were raisedfor one week and assigned randomly into seven groups of six rats for intragastric administration of: model, intact, reference product of 30-mg enteric-coated capsules of LSZ (2.7 mg/kg), LSZ (2.7 mg/kg), test product compound of 30-mg capsule of LSZ (LSZ:NaHCO_3_): (1.35 mg/49 mg), (2.7 mg/99 mg) and (5.4 mg/198 mg) for compound LSZ-L, compound LSZ-M and compound LSZ-H groups, respectively. All animals except the intact group had their stomach injected with glacial acetic acid (0.03 mL) to induce gastric ulcers according to a previously described method [19]. All groups received oral administration of the relevant medication for seven consecutive days. The model and intact group rats were given the same volume of saline orally for seven consecutive days. After the penultimate administration, rats were starved for 24 h. Animals were sacrificed under anesthesia and their stomachs were removed for analysis.

#### 2.2.3. Macroscopic Gastric Ulcer Assay

The gastric ulcer index (UI) was the total area of ulcer. The gastroprotection (%) was calculated and used for macroscopic gastric ulcer assay [20].

#### 2.2.4. Histopathological Observation

Cross-sections with a thickness of 4 µm were then papered for hematoxylin and eosin (H&E) staining. All specimens were assessed under a light microscopy [21].

#### 2.2.5. Immunohistochemistry Evaluation for VEGF and COX-2

Gastric tissues were post fixed in 4% paraformaldehyde overnight at 4 °C before routine paraffin sections (4 μm) were prepared for immunohistochemical staining. Paraffin-embedded slices of tissues were incubated overnight at 4 °C with primary rabbit anti-VEGF and anti-COX-2 antibody (1:200, Thermo). After rinsing, sections were incubated in biotinylated goat anti-rabbit IgG (1:200, Maixin) at 37 °C for 30 min and were subsequently incubated in streptavidin-peroxidase conjugate (1:200, Maixin) at 37 °C for 30 min. The sections were developed with 3,3′-diaminobenzidine (DAB) in a chromogen solution and counter-stained with hematoxylin. Images were obtained using an Olympus BX-61microscope (Tokyo, Japan).

#### 2.2.6. Western Blotting Determination for VEGF and COX-2

The homogenate was centrifuged at a force of 12,000 g for 30 min at 4 °C and protein qualification was carried out using a BCA kit. The total protein extract (25 μg per lane) was separated on SDS-polyacrylamide gels and transferred onto PVDF membranes. Nonspecific binding sites on the membrane were blocked by PBS containing 0.1% Tween-20 with 5% bovine serum albumin, which was followed by incubation with rabbit polyclonal primary antibodies against VEGF and COX-2 and β-actin overnight at 4 °C. Immunoreactive bands were visualized using the enhanced chemiluminescent kit. Bands were measured using the Quality One analysis software and the density of each band was normalized to β-actin.

#### 2.2.7. Measurement of SOD, MDA and H^+^-K^+^-ATP Enzymes Activity Levels

The activities of GSH, SOD and H^+^-K^+^-ATP enzymes were tested by using commercially available kits of the compound LSZ capsule on glacial acetic acid-induced gastric ulcer. Lipid peroxidation was determined by assessing the level of thiobarbituric acid reactive substances (TBARS), which was measured as MDA according to the manufacturer’s instructions [22].

### 2.3. Pharmacokinetic Study in Dogs

#### 2.3.1. Animals

Six beagle dogs (three males, Shenyang KangPing Institute of Animal Laboratory, SCXK [Liao] 2014-0003, Shenyang, China) were tested by four cycle tests. Each experimental cleaning period was two weeks. The first three cycle tests consisted of the oral administration of the test product compound of a 30-mg capsule of LSZ (LSZ: NaHCO_3_, 30 mg/1.1 g) at concentrations of 0.5 mg·kg^−1^, 1 mg·kg^−1^ and 2 mg·kg^−1^. The last cycle test was an oral administration of the reference product of 30-mg enteric-coated capsules of LSZ at 1 mg·kg^−1^. 

The experimental protocol was approved by the Animal Ethics Committee of China Medical University (Permit number: SYK2017-0022), and all animal studies were carried out in accordance with the Guidelines for Animal Experimentation [23].

#### 2.3.2. Instruments

The samples were analyzed on a 3500 MS/MS Applied AB Sciex coupled with an Aglient UPLC 1290 system. An Aglient ZORBAX Eclipse Plus C18 (2.1 mm × 100 mm, 1.8 μm) at the temperature of 30 °C was used for separation. The mobile phase consisted of methanol and 0.1% formic acid water (60:40 *v*/*v*) and was delivered at a flow rate of 0.3 mL·min^−1^. The mass spectrometer was operated in the positive ion mode with a Turbo Ion Spray source. Table 1 shows the optimized MRM parameters.

#### 2.3.3. Solution Preparation

The standard stock solution (1000 μg·mL^−1^) of the LSZ and omeprazole (IS) were prepared by methanol/water (60:40, *v*/*v*).

Calibration standard solutions were prepared by spiking these working solutions into drug-free rat plasma. LSZ was set at concentrations of 5, 10, 25, 125, 625, 1250 and 2500 ng·mL^−1^. An internal standard working solution (4000 ng·mL^−1^) was also prepared. 

#### 2.3.4. Sample Preparation

Plasma samples with a volume of 100 μL, 10 μL IS solution and 10 μL methanol/water were transferred together into a 10-mL centrifuge tube. Vortex shaking was conducted for 30 s before 1 mL of ethyl acetate was added. The analytes were extracted by vortexing for 5 min and shaking for 5 min. The samples were then centrifuged at a force of 3000× *g* for 5 min. The organic layer was transferred and evaporated until it was dry by nitrogen stream. Finally, the dried extract was reconstituted in 100 μL of the solvent (methanol/water [60:40, *v*/*v*]) before it underwent UPLC-MS/MS analysis.

#### 2.3.5. Method Validation

The method was fully validated in accordance with the US Food and Drug Administration (FDA) guidelines [24,25]. Specificity, linearity, LLOQ, precision, accuracy, recovery, matrix effect and stability were all experimentally investigated.

#### 2.3.6. Method Application

The six beagle dogs were orally administered with 0.5 mg·kg^−1^, 1 mg·kg^−1^ and 2 mg·kg^−1^ of 30-mg capsule of compound LSZ and 1 mg·kg^−1^ of the reference product (30-mg enteric-coated capsule of LSZ). Blood samples (1 mL) were drawn via the foreleg vein into tubes from each dog before administration and 0.083, 0.17, 0.33, 0.5, 0.67, 1.0, 1.5, 2.0, 3.0, 4.0, 5.0, 6.0, 7.0, 8.0 and 10 h after administration.

### 2.4. Data Analysis and Statistics

The pharmacokinetic parameters of the LSZ were calculated by the DAS 2.1 software package [26]. Statistical comparisons between the test and reference product groups were performed with SPSS 16.0, with *p* < 0.05 considered to be statistically significant. The pharmacodynamic data were statistically analyzed in the same way.

## 3. Results

### 3.1. Pharmacodynamic Studies in Rats

#### 3.1.1. Macroscopic Evaluation of Compound LSZ Capsule against Glacial Acetic-Acid-Induced Gastric Ulcer

The results of the macroscopic evaluation of the compound LSZ capsule against glacial acetic-acid-induced gastric ulcers are shown in Figure 2. The intact group showed no stomach injuries. Numerous hemorrhagic red bands of different sizes were noticeably observed in gastric mucosa of the model group. Macroscopic images revealed that treatment with LSZ significantly reduced gastric lesions compared to the model group, with the compound LSZ-M being the most effective treatment. The UI and gastroprotection (%) were quantified, as shown in Table 2, with compound LSZ-M having remarkable gastroprotective effects compared with the model and other LSZ groups.

#### 3.1.2. Histological Evaluation of Compound LSZ Capsule against Glacial Acetic-Acid-Induced Gastric Ulcer

The results of the histological evaluation of the glacial acetic acid-induced gastric ulcers after administration of the compound LSZ capsule are shown in Figure 3. In the intact group, the lining of the gastric sinus was evenly distributed and the mucosal muscle was normal with no inflammatory cells or bleeding present. The glacial acetic-acid-induced model group had severe damage to the gastric submucosa edema and epithelial cell loss. Treatment with LSZ had a protective effect on gastric mucosa, which was shown by the reduction of gastric submucosal edema and epithelial cell loss as compared to the model group. The most important result was that the compound LSZ-M was the most effective treatment of the studied LSZ groups.

#### 3.1.3. Immunohistochemistry Evaluation for VEGF and COX-2

Vascular endothelial growth factor (VEGF) can be applied to vascular endothelial cells, which promotes angiogenesis, maintains normal blood vessels and the integrity and increases vascular permeability. Thus, VEGF plays an important role in tissue repair and angiogenesis. Cyclic oxygenase 2 (COX-2) can catalyze the synthesis of prostaglandin (PGE2), which has multiple mucosal protection effects and could regulate the contraction and relaxation of gastric mucosal microvasculature, repairing mucosal damage and promoting the healing of ulcers. Immunohistochemistry results showed the effect of compound LSZ capsule on the expression of VEGF and COX-2. In Figure 4, the LSZ group displayed increased expression of VEGF and COX-2 (*p* < 0.01), while the compound LSZ-M group displayed significantly raised expression of VEGF and COX-2 compared to the other LSZ groups.

#### 3.1.4. Western Blotting Determination for VEGF and COX-2

In order to further verify whether the compound LSZ capsule has a better gastroprotective effect, we also assayed the expression of VEGF and COX-2 proteins by Western blot. As shown in Figure 5, the model group decreased expression of VEGF and COX-2. On the contrary, LSZ could significantly increase the expression of VEGF and COX-2 proteins (*p* < 0.01). The compound LSZ-M group was again the best treatment group.

#### 3.1.5. Measurement of SOD, MDA and H^+^-K^+^-ATP Enzymes Activity Levels

SOD activity is one of the important catalysis and oxygen-removalenzymes. Furthermore, it is an important contributor to the protection of the gastric mucosa as it cannot only prevent gastric mucosal injury but can also promote the repair of damage. MDA is a metabolite of lipid peroxide, which has strong biological toxicity and can aggravate the formation of ulcers. LSZ is transformed into an active body structure under acidic conditions, which can be combined with the SH group of H^+^-K^+^-ATP to inhibit proton pump activity, thus inhibiting gastric acid secretion. As shown in Table 3, LSZ groups increased the SOD levels in gastric (*p* < 0.05) and also reduced MDA and H^+^-K^+^-ATP enzyme activity levels (*p* < 0.05). The compound LSZ-M group showed a marked reversalin the SOD, MDA and H^+^-K^+^-ATP enzyme activity levels compared to the intact group, which demonstrated that this was the best treatment group.

### 3.2. Pharmacokinetic Study in Dogs

#### 3.2.1. Method Validation

No endogenous interference was observed in the typical chromatograms of blank dog plasma as well asblank dog plasma spiked with LSZ and IS. The dog plasma samples 2 h after the administration of LSZ at a dose of 1.0 mg·kg^−1^ are shown in Figure 6.

Linearity and LLOQ calibration curves were linear over the concentration range of 5.0–2500 ng·mL^−1^, with the corresponding linear regression equation of *y* = 0.00222*x* + 0.02517 (*n* = 3, *r* = 0.9975). The lower limit of the quantification of LSZ was 5 ng·mL^−1^. Precision, accuracy, recovery and matrix effect results indicated that the method had good precision and accuracy. The recovery was 84.98 ± 6.75% for the IS. The matrix effect indicated that no significant matrix effect was observed for the LSZ and IS. The results are shown in Table 4.

The stability results indicate that the analyte can be considered to be stable. The results are shown in Table 5.

#### 3.2.2. Method Application

The UPLC-MS/MS analysis of LSZ in beagle dog plasma has been previously demonstrated [25]. The mean plasma concentration versus time profiles of test product LSZ for three dose groups after the oral administration of LSZ to individual dogs (*n* = 6) are illustrated in Figure 7. The mean plasma concentration versus time profiles of test product and reference product LSZ at same dose groups was illustrated in Figure 8. Furthermore, the main pharmacokinetic parameters are summarized in Table 6. The results demonstrate that plasma concentrations increased rapidly and reached the mean *C*_max_ concentrations of 710.6, 1390.7 and 2067.2 ng·mL^−1^ for three dose groups and the *T*_max_ were 0.67, 0.81 and 0.56 h. The *C*_max_ and AUC tended to increase in proportion to the dose with the linear regression of *C*_max_ = 871.8 Dose + 372.35 (*r*^2^ = 0.9637), AUC_0–t_ = 1718 Dose + 992.5 (*r*^2^ = 0.9611) and AUC_0–∞_ = 1408 Dose + 1794 (*r*^2^ = 0.9982) without significant differences for plasma clearance (Cl), which indicated that linear pharmacokinetic properties of LSZ were observed in the given three dosage groups. There was a significant difference (*p* < 0.05) in the pharmacokinetic parameter of *T*_max_ between two groups. The *T*_max_ value was observed to be significantly reduced in the test product group compared to the reference product group. The results showed the advantage of the new type of rapidly absorbed compound LSZ capsule.

## 4. Discussion

The cause of gastric ulcers is complex, which is generally believed to cause gastric mucosal injuries for various reasons, resulting in decreased resistance to gastric acid and pepsin. Thus, this causes the symptoms of ulcer. The pathological model of ulcers can be induced by chemical and physical stimulation as well as in astress state. The model of gastric ulcers induced by glacial acetic acid is simple and has a high incidence of ulcers compared to the other gastric ulcer model. Therefore, we used this model to study whether compound LSZ capsules can prevent gastric ulcers.

VEGF can be applied in vascular endothelial cells and promotes angiogenesis, which plays an important role in tissue repair. COX-2 can catalyze the synthesis of PGE2 and promotes the healing of ulcers. Therefore, an increase in VEGF and COX-2 levels could be useful for protection against gastric ulcers. SOD activity is one of the important catalysis and oxygen-removal enzymes. SOD can not only prevent gastric mucosal injury, but can also promote the repair of damage. MDA is a metabolite of lipid peroxide, which has strong biological toxicity and can aggravate the formation of ulcers. The increase and decrease in MDA content represents the strength of lipid peroxidation and indirectly reflects the degree of cell damage [27]. LSZ is transformed into an active body structure under acidic conditions, which can be combined with the SH group of H^+^-K^+^-ATP to inhibit proton pump activity, thus inhibiting gastric acid secretion. Therefore, a decrease in the activity level of MDA and H^+^-K^+^-ATP enzymes could be useful for the protection of gastric ulcers.

Our result is consistent with the idea that these oxidative stress and inflammatory factors could be used as a measure for the treatment of gastric ulcers. The results showed that the LSZ groups can promote the healing of gastric ulcers by increasing the expression of mucosal VEGF and COX-2, enhancing SOD activity and decreasing MDA activity and H^+^-K^+^-ATP activity. Additionally, the compound LSZ-M group displayed markedly reversed activity levels of SOD, MDA and H^+^-K^+^-ATP enzymes compared to the intact group, which demonstrated that it was the best treatment group. Therefore, we confirmed the compound LSZ 30-mg capsule (LSZ: NaHCO_3_): (30 mg/1.1 g) as the optimal dose for pharmacokinetics studies in dogs.

The reason that the new type of compound capsule has an advantage is that the acid neutralization agent NaHCO_3_ was added into the prescription. First, a neutral environment in the stomach was created after oral administration of the drug, after which it was able to protect the stability of LSZ. The LSZ was released and absorbed in the stomach, which could neutralize hydrochloric acid in gastric juice and quickly inhibit gastric acid secretion. Lastly, it was shown that the merit of compound LSZ capsule was its rapid absorption compared with the enteric capsule. This study could be helpful for improving the preclinical therapeutic efficacy of LSZ.

## 5. Conclusions

The present study demonstrated that compound lansoprazole capsule significantly reduced gastric ulcer through regulating anti-inflammatory cytokines and oxidative stress. It was able to enhance the expression of the antioxidant enzyme SOD and suppress lipid peroxidation, which was indicated by the reduction of MDA and H^+^-K^+^-ATP activity, as well as to promoting the healing of gastric ulcer by increasing the expression of mucosal VEGF and COX-2. Additionally, an efficient UPLC-MS/MS method for the determination of the new type of rapidly absorbed compound LSZ capsule in the dog plasma has been developed and validated. The compound LSZ capsule had the advantage of rapid absorption. This study could be beneficial for the application of LSZ in preclinical therapy.

## Figures and Tables

**Figure 1 pharmaceutics-11-00049-f001:**
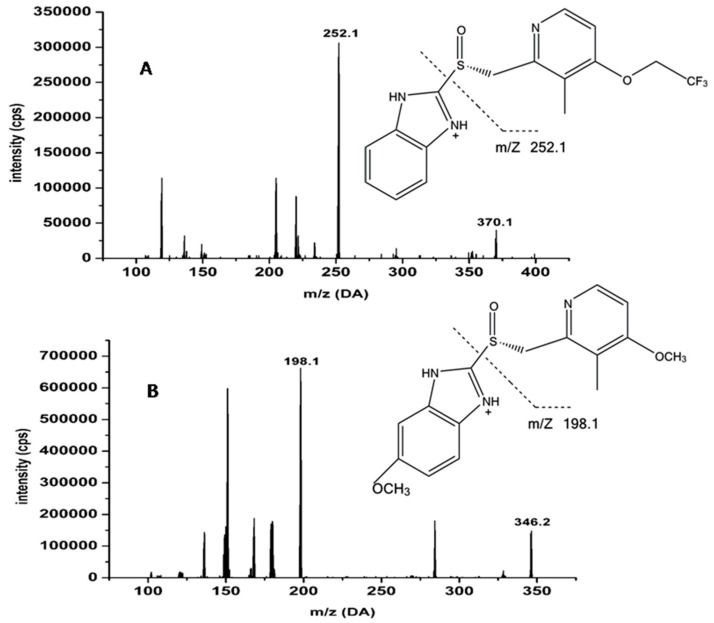
Chemical structures and full scan product ion of precursor ions of lansoprazole (**A**) and omeprazole (**B**; IS).

**Figure 2 pharmaceutics-11-00049-f002:**
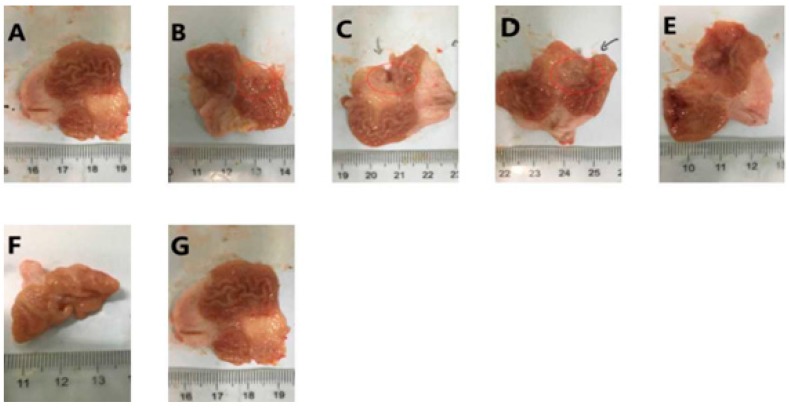
Effects of compound LSZ capsule on the macroscopic appearance of rats’ gastric mucosa, which has been damaged (*n* = 6). Seven groups: (**A**) Intact; (**B**) Model; (**C**) reference product of 30-mg enteric-coated capsules of LSZ (2.7 mg/kg); (**D**) LSZ (2.7 mg/kg); (**E**–**G**) test product of 30-mg capsules of compound LSZ (LSZ:NaHCO_3_): (1.35 mg/49 mg), (2.7 mg/99 mg) and (5.4 mg/198 mg) for compound LSZ-L, compound LSZ-M and compound LSZ-H groups, respectively.

**Figure 3 pharmaceutics-11-00049-f003:**
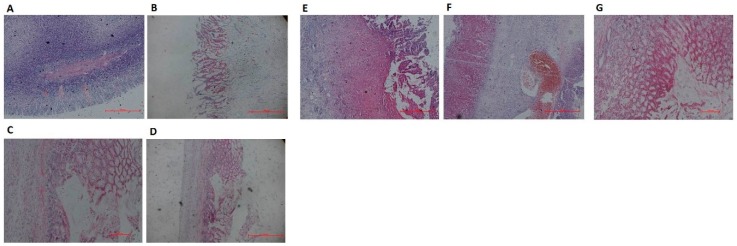
Effects of compound LSZ capsule on histopathological lesions of gastric mucosa in rats (*n* = 6). **A**, Intact; **B**, Model; **C**, reference product of 30-mg enteric-coated capsules of LSZ (2.7 mg/kg); **D**, LSZ (2.7 mg/kg); **E**, **F**, **G**, test product of 30-mg capsules of compound LSZ (LSZ:NaHCO_3_): (1.35 mg/49 mg), (2.7 mg/99 mg) and (5.4 mg/198 mg) for compound LSZ-L, compound LSZ-M and compound LSZ-H groups, respectively.

**Figure 4 pharmaceutics-11-00049-f004:**
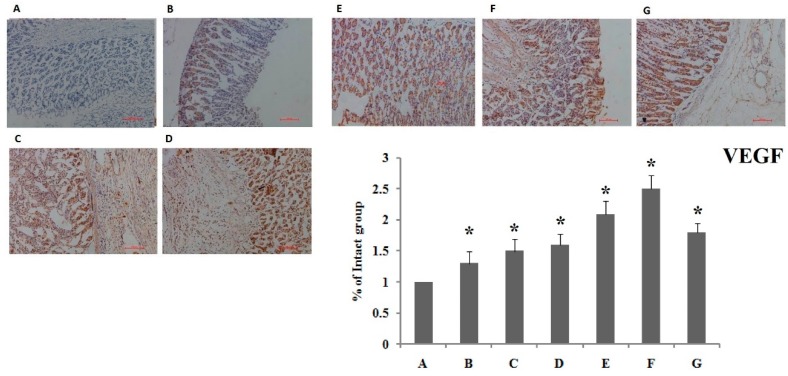
Immunohistochemistry results showing the effect of compound LSZ capsule on the expression of VEGF and COX-2 (*n* = 6). **A**, Intact; **B**, Model; **C**, reference product of 30-mg enteric-coated capsules of LSZ (2.7 mg/kg); **D**, LSZ (2.7 mg/kg); **E**, **F**, **G**, test product of 30-mg capsules of compound LSZ (LSZ:NaHCO_3_): (1.35 mg/49 mg), (2.7 mg/99 mg) and (5.4 mg/198 mg) for compound LSZ-L, compound LSZ-M and compound LSZ-H groups, respectively. * *p* < 0.05 compared with intact group.

**Figure 5 pharmaceutics-11-00049-f005:**
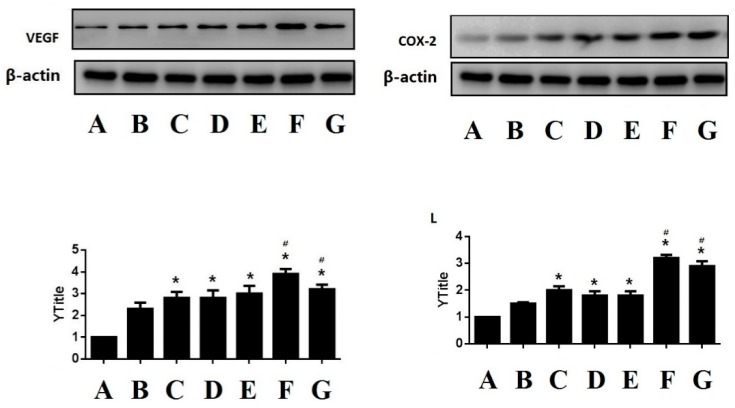
Western blot result showed the effect of compound LSZ capsule on the expression of VEGF and COX-2 (*n* = 6). (**A**) Intact; (**B**) Model; (**C**) reference product of 30-mg enteric-coated capsules of LSZ (2.7 mg/kg); (**D**) LSZ (2.7 mg/kg); (**E**–**G**) test product of 30-mg capsules of compound LSZ (LSZ:NaHCO_3_): (1.35 mg/49 mg), (2.7 mg/99 mg) and (5.4 mg/198 mg) for compound LSZ-L, compound LSZ-M and compound LSZ-H groups, respectively.* *p* < 0.05 compared with intact group.

**Figure 6 pharmaceutics-11-00049-f006:**
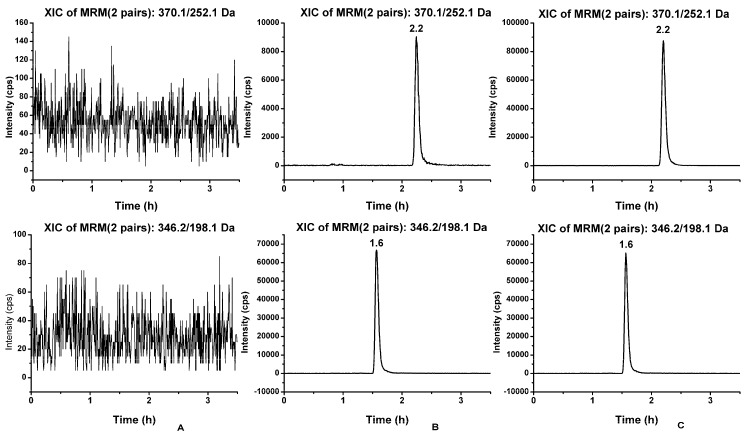
Typical chromatograms of (**A**) blank dog plasma; (**B**) blank dog plasma spiked with lansoprazole and IS at LLOQ; and (**C**) dog plasma sample 2 h after administration of lansoprazole at a dose of 1.0 mg·kg^−1^. Representative MRM chromatograms of lansoprazole (*t*_R_ = 2.2) and omeprazole (*t*_R_ = 1.6).

**Figure 7 pharmaceutics-11-00049-f007:**
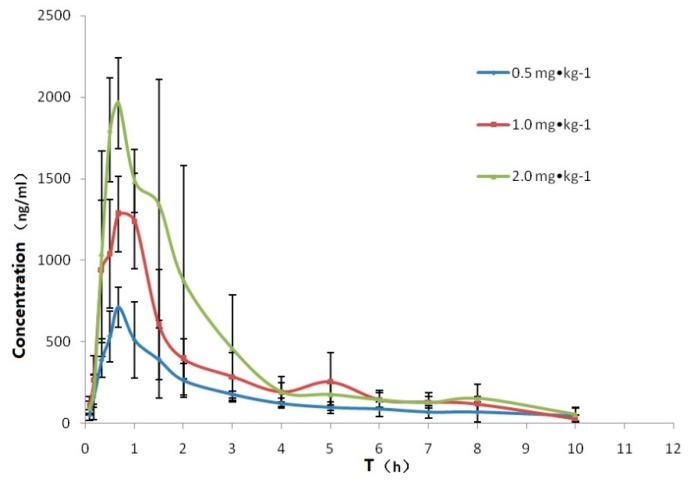
Plasma concentration–time curves for lansoprazole in three dose groups. Each point represents the mean ± S.D (*n* = 6).

**Figure 8 pharmaceutics-11-00049-f008:**
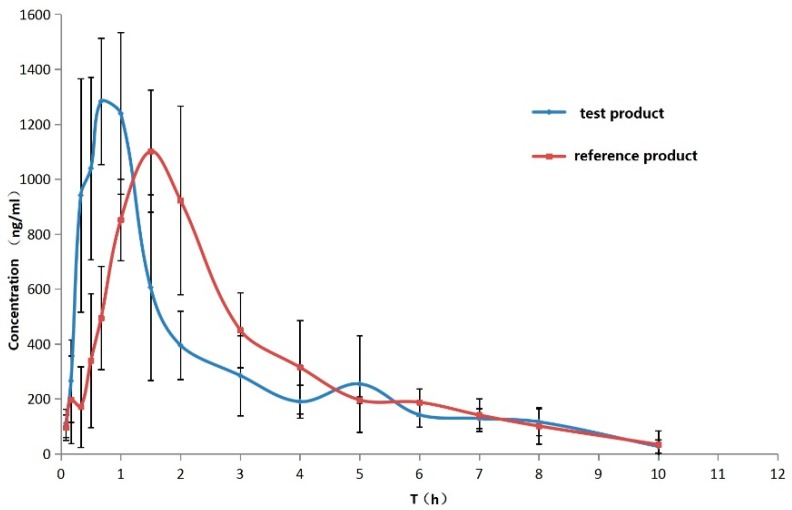
Plasma concentration–time curves for lansoprazole in test product and reference product groups at the dose of 1.0 mg·kg^−1^. Each point represents the mean ± S.D. (*n* = 6).

**Table 1 pharmaceutics-11-00049-t001:** Optimized multiple-reaction-monitoring (MRM) parameters for lansoprazole and omeprazole (IS).

Analytes	Q1 (amu)	Q3 (amu)	DP (V)	CE (eV)	*t*_R_ (min)
lansoprazole	370.1	252.1	70	23	2.2
omeprazole (IS)	346.2	198.1	70	30	1.6

**Table 2 pharmaceutics-11-00049-t002:** The UI and gastroprotection (%) of the compound LSZ capsule with * *p* < 0.05, ** *p* < 0.01 compared with model group (mean ± S D; *n* = 6).

Groups	UI	Gastroprotection (%)
Intact	0	0
Model	39.40 ± 7.73	0
reference product	20.60 ± 5.29 *	48.5
LSZ (2.7 mg/kg)	28.30 ± 7.29 *	29.8
compound LSZ-L	25.20 ± 4.75 *	37.0
compound LSZ-M	0.70 ± 0.29 **	98.2
compound LSZ-H	8.50 ± 5.32 *	79.3

**Table 3 pharmaceutics-11-00049-t003:** Effect of compound LSZ capsuleon the activity levels of SOD, MDA and H^+^-K^+^-ATP enzymes. * *p* < 0.05 and ** *p* < 0.01 compared with model group (mean ± SD; *n* = 6).

Groups	SOD (μmol*g^−1^ Pro)	MDA (μmol*g^−1^ Pro)	H^+^-K^+^-ATP enzyme activity (mmol/g h)
Intact	201.23 ± 16.50	73.08 ± 8.76	7.13 ± 0.48
Model	156.12 ± 12.00	97.57 ± 7.77	8.65 ± 0.54
reference product	166.96 ± 12.53 *	72.65 ± 8.98 *	7.82 ± 0.54 *
LSZ (2.7 mg/kg)	170.12 ± 13.19 *	75.32 ± 10.77 *	7.91 ± 0.76 *
compound LSZ-L	170.00 ± 10.08 *	87.00 ± 9.67 *	7.83 ± 0.84 *
compound LSZ-M	187.21 ± 8.20 *	40.05 ± 6.93 **	7.20 ± 0.93 *
compound LSZ-H	172.01 ± 9.40 *	54.58 ± 7.73 *	7.52 ± 0.39 *

**Table 4 pharmaceutics-11-00049-t004:** Summary of accuracy, precision, recovery and matrix effects of the lansoprazole in dog plasma (*n* = 6).

Analytes	Concentration	Intra-Day	Inter-Day	Accuracy	Recovery	Matrix Effect
	(ng·mL^−1^)	R.S.D. (%)	R.S.D. (%)	(RE%)	(%, mean ± SD)	(%, mean ± SD)
lansoprazole	10	9.4	12.4	−1.8	83.69 ± 3.80	82.68 ± 4.81
	125	7.1	2.0	1.1	78.35 ± 5.24	90.53 ± 1.20
	2000	5.1	13.3	−1.2	94.70 ± 3.39	82.13 ± 0.34

**Table 5 pharmaceutics-11-00049-t005:** Stability of the lansoprazole in dog plasma (*n* = 3).

Analytes	Concentration (ng·mL^−1^)	24 h, Room Temperature		45 Days, −20 °C		3 Freeze-Thaw Cycles		10 h, 4 °C	
lansoprazole	10	10.00	12.08	9.11	4.06	9.64	10.03	9.35	7.2
	125	132.80	5.28	123.82	11.27	119.79	8.83	124.51	4.3
	2000	2024.38	8.40	1853.17	2.92	2128.46	4.07	2088.22	2.9

**Table 6 pharmaceutics-11-00049-t006:** Main plasma pharmacokinetic parameters of the three different doses of the lansoprazole test product and reference product, * *p* < 0.05 compared with *T_max_* (mean ± SD; *n* = 6).

Parameters	0.5 mg·kg^−1^	1.0 mg·kg^−1^	2.0 mg·kg^−1^	1.0 mg·kg^−1^ (RP)
*AUC*_0–t_ (μgh/L)	1652 ± 589.7	3010 ± 536.8	4330 ± 1024	3314 ± 307.4
*AUC*_0–∞_ (μgh/L)	2533 ± 1695	3151 ± 569.4	4630 ± 938.0	3487 ± 242.7
T_1/2_ (h)	1.66 ± 1.16	1.87 ± 1.12	2.76 ± 2.03	1.7 ± 1.22
*T_max_* (h)	0.67 ± 0	0.81 ± 0.22	0.56 ± 0.14	1.67 ± 0.23 *
*C_max_* (ng/L)	710.6 ± 123.7	1391 ± 198.3	2067 ± 266.4	1197 ± 185.7
MRT (h)	2.83 ± 0.37	2.72 ± 0.50	2.5 ± 0.41	3.01 ± 0.23
CL (L/h/kg)	2.72 ± 1.39	3.27 ± 0.63	4.46 ± 0.85	2.88 ± 0.22
Vd (L/kg)	23.14 ± 18.10	8.58 ± 4.62	17.54 ± 12.94	7.04 ± 4.94

*AUC*_0–t_, *AUC*_0–∞_, area under the plasma concentration–time curve from time 0 to *t*, 0 to ∞; MRT, the sum means absorption and mean residence time; *T*_max_, time to reach the maximum plasma concentration; *C*_max_, peak plasma; *t*_1/2_, terminal elimination half-life.

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
