# Peer review of "Pharmacodynamics and Pharmacokinetics of a New Type of Compound Lansoprazole Capsule in Gastric Ulcer Rats and Beagle Dogs: Importance of Adjusting Oxidative Stress and Inflammation"

_pharmaceutics, 2019, doi:10.3390/pharmaceutics11020049_

Reviewer 1 Report

Although most of the issues in the prior submission has been addressed, the manuscript lacks coherency and has multiple grammatical mistakes. Most of the sentences are difficult to understand. Thus, I'd recommend the authors to submit the manuscript for thorough English language checking by native English speaker with scientific background. The manuscript could be accepted for publication after extensive language editing. 

Author Response

Response to Reviewer #1:

Reply: Thanks for your helpful comments.The manuscript has undergone English language editing by MDPI. The text has been checked for correct use of grammar and common technical terms, and edited to a level suitable for reporting research in a scholarly journal.

Reviewer 2 Report

The manuscript as presented describes the preparation and screening of a new formulation which contains lansoprazole. The formulation has been assessed for its ability to suppress gastric ulcer formation and progression, the data for which is presented and discussed. It is clear that some amendments have been made to the manuscript, as highlighted, but the readability of the manuscript is blighted by errors in English language and style which must be addressed and corrected before the manuscript is accepted for publication. As may well have been recommended previously, the manuscript will benefit greatly from being proof read by someone for whom English is their first language.

Author Response

Response to Reviewer #2:

Reply: Thanks for your helpful comments.The manuscript has undergone English language editing by MDPI. The text has been checked for correct use of grammar and common technical terms, and edited to a level suitable for reporting research in a scholarly journal.

Reviewer 3 Report

Accepted in current form. Author(s) answered all the queries appropriately. 

Author Response

Response to Reviewer #3:

Reply: Thanks for your helpful comments.The manuscript has undergone English language editing by MDPI. The text has been checked for correct use of grammar and common technical terms, and edited to a level suitable for reporting research in a scholarly journal.